# Matrix method and the suppression of Runge's phenomenon

Shui-Fa Shen[1,2,3],[*] Wei-Liang Qian[4,5],[†] Jie Zhang[6],[‡]
Yu Pan[7],[§] Yu-Peng Yan[8,9],[¶] and Cheng-Gang Shao[10][**]

[1] *School of intelligent manufacturing, Zhejiang Guangsha Vocational and
Technical University of Construction, 322100, Jinhua, Zhejiang, China*

[2] *School of Electronic, Electrical Engineering and Physics,
Fujian University of Technology, 350118, Fuzhou, Fujian, China*

[3] *Hefei Institutes of Physical Science, Chinese Academy of Sciences, 230031, Hefei, Anhui, China*

[4] *Escola de Engenharia de Lorena, Universidade de São Paulo, 12602-810, Lorena, SP, Brazil*

[5] *Center for Gravitation and Cosmology,
School of Physical Science and Technology,
Yangzhou University, 225002, Yangzhou, Jiangsu, China*

[6] *College of Physics and Electronic Engineering,
Qilu Normal University, 250300, Jinan, Shangdong, China*

[7] *College of Science, Chongqing University of Posts and Telecommunications, 400065, Chongqing, China*

[8] *School of Physics, Suranaree University of Technology, Nakhon Ratchasima 30000, Thailand*

[9] *Thailand Center of Excellence in Physics,
Commission on Higher Education, Ratchathewi, Bangkok 10400, Thailand and*

[10] *MOE Key Laboratory of Fundamental Physical Quantities Measurement,
Hubei Key Laboratory of Gravitation and Quantum Physics, PGMF, and School of Physics,
Huazhong University of Science and Technology, 430074, Wuhan, Hubei, China*
(Dated: Sept. 9th, 2023)

Higher-degree polynomial interpolations carried out on uniformly distributed nodes are often plagued by *overfitting*, known as Runge's phenomenon. This work investigates Runge's phenomenon and its suppression in various versions of the matrix method for black hole quasinormal modes. It is shown that an appropriate choice of boundary conditions gives rise to desirable suppression of oscillations associated with the increasing Lebesgue constant. For the case of discontinuous effective potentials, where the application of the above boundary condition is not feasible, the recently proposed scheme with delimited expansion domain also leads to satisfactory results. The onset of Runge's phenomenon and its effective suppression are demonstrated by evaluating the relevant waveforms. Furthermore, we argue that both scenarios are either closely related to or practical imitations of the Chebyshev grid. The implications of the present study are also addressed.

---

[*] E-mail: shenshuifa2006@163.com

[†] E-mail: wlqian@usp.br (corresponding author)

## I. INTRODUCTION

The black hole quasinormal modes (QNMs) carry essential information on one of the most crucial predictions of Einstein's general relativity. These distinctive dissipative oscillations bear intrinsic properties of the peculiar spacetime region [1–3]. From an observational perspective, such a temporal profile may manifest as the ringdown phase of a merger through which the black hole is formed. The associated signals are physically straightforward and mathematically "clean" compared to those emanating from other stages of the merger process. Meanwhile, the direct detection of gravitational waves (GWs)[4–7] has provided unique insights and raised further expectations, widely recognized as heralding a new era of GW astronomy. Specifically, it has been speculated that the strength of these signals is feasible for space-borne GW projects currently under development, such as LISA[8], TianQin [9], and Taiji [10].

Besides its experimental relevance, black hole QNM is also an intriguing theoretical topic in its own right. The QNMs are characterized by a spectrum of complex frequencies, subject to the in-going and out-going boundary conditions defined at the horizon and the outer spatial boundary, respectively. As a scattering problem, these discrete modes emerge when the transmission or reflection coefficient becomes divergent. Schutz and Will argued that specific resonance might occur if the waveform frequency is numerically close to the peak of the effective potential in the complex plane, especially when the transmission and reflection coefficients are of similar magnitude. These arguments gave rise to the well-known WKB approximation [11, 12]. Ferrari and Mashhoon [13] transformed the problem into a bound state by extending the spatial wave function domain onto the imaginary axis through analytic continuation. One of the most accurate approaches, known as the continued fraction method, was proposed by Leaver [14]. To solve for the complex frequencies, the wave function is expanded near the horizon, and its asymptotic behavior is stripped away, reminiscent of the problem of the hydrogen spectrum in quantum mechanics. A similar strategy was employed in the matrix method [15–21], proposed by some of us, where a matrix equation for the complex quasinormal frequencies is solved. Additionally, the hyperboloidal approach [22] reformulated the task as an eigenvalue equation for a non-selfadjoint operator by replacing spacelike infinity in the original problem with null infinity. Leaver treated the waveform in terms of Fourier spectrum decomposition [23], a technique further explored by Nollert and Schmidt [24] using Laplace transforms. This last framework identifies the QNMs as the poles of the corresponding frequency domain Green's function.

On the practical side, much effort has been devoted to extracting information from the empirical waveforms, known as the black hole spectroscopy [25–30]. Moreover, a black hole or other compact object is always merged in a realistic environment, and the resulting spacetime deviates from that entirely governed by a mathematical solution in the vacuum. Subsequently, the emanated GWs might deviate substantially from an isolated object in the ideal scenario. The latter leads to the notion of *dirty* black holes [31–34]. Many pertinent studies have been carried out in this direction. The scalar QNMs of dirty black holes were first explored by Leung *et al.* using the generalized logarithmic perturbation theory. A comprehensive analysis was performed by Barausse *et al.*, where the time evolution of small perturbations about a central Schwarzschild black hole [34] was investigated. It was observed that the evaluated QNMs might be substantially different from those of an ideal black hole. Nonetheless, it was concluded that the astrophysical environment would not significantly affect the black hole spectroscopy if an appropriate waveform template

‡ E-mail: zhangjie@qlnu.edu.cn

§ E-mail: panyu@cqupt.edu.cn

¶ E-mail: yupeng@g.sut.ac.th

** E-mail: cgshao@hust.edu.cn

was adopted. The above studies have partly promoted the recent developments on QNM instability [35–37]. In particular, for a non-Hamiltonian system, the notion of pseudospectrum has been employed to understand the system's instability that cannot be captured by linear stability analysis. In the context of black hole perturbation theory, the study was initiated by Nollert [38] and followed up by Nollert and Price [39], Daghigh *et al.* [40], and some of us [41]. It was demonstrated that the high-overtone modes of the QNM spectrum are unstable when triggered by ultraviolet perturbations. This conclusion undermines the educated guess that the QNM spectrum is not expected to be significantly different once a reasonable approximation for the effective potential is adopted. In fact, the presence of an insignificant discontinuity might have a drastic impact on the asymptotical behavior of the QNM spectrum [41]. From a general perspective, Jaramillo *et al.* [35–37] demonstrated that the boundary of the pseudospectrum migrates toward the real frequency axis as a result of randomized perturbations. These studies indicate a universal instability of the high-overtone modes and their potential implications on GW astronomy. Recently, Cheung *et al.* [42] further showed that even the fundamental mode can be destabilized under generic perturbations. The above studies call for numerical results with unprecedented precision. In particular, it is meaningful to further study the effect of the relevant perturbations on the non-selfadjoint operator [35] in the context of black hole physics. Among others, effective potentials with discontinuity might play an exciting role as a mathematically simple and physically relevant model.

The matrix method [15–21] is an approach that reformulates the QNM problem into an algebraic nonlinear equation for the complex frequencies. Compared with the continued fraction method, the main difference resides in the choice of the grid points where the expansion of the waveform is performed [15]. Besides the metrics with spherical symmetry [16], the method can be applied to black hole spacetimes with axial symmetry [17] and system composed of coupled degrees of freedom [43]. The approach is shown to be effective in dealing with different boundary conditions [18] and dynamic black holes [19]. More recently, the original method was generalized [20] to handle effective potentials containing discontinuity and pushed to the higher orders [21]. The method matrix method is shown to offer reasonable accuracy as well as efficiency and has been adopted in various studies [40, 44–56]. Nonetheless, it also entails some drawbacks, specifically those associated with the equispaced interpolation points implemented for most applications. It is well-known that polynomial interpolation based on a uniformed grid is often liable to Runge's phenomenon, characterized by significant oscillations at the edges of the relevant interval, closely related to an increasing Lebesgue constant [57]. In other words, even though the uniform convergence over the interval in question is provided, an interpolation of a higher degree does not necessarily guarantee an improved accuracy, similar to the Gibbs phenomenon in Fourier series approximations.

The present study is primarily motivated by the above considerations. We investigate the potential Runge's phenomenon in various versions of the matrix method for black hole QNMs. It is shown that an appropriate choice of boundary conditions can effectively suppress or delay the onset of the phenomenon. Implementing a delimited expansion domain might also lead to reasonable results when such boundary conditions are not feasible. Moreover, we note that the recent results on the instability of the fundamental mode [42] invite further studies to explore the black hole QNMs with unprecedented high precision. In this regard, such an analysis is also physically meaningful.

The remainder of the paper is organized as follows. In the following section, we briefly summarize various versions of the matrix method. After briefly reviewing Runge's phenomenon, we demonstrate its onset in various scenarios in Sec. III. The deviation of the interpolant is estimated in terms of the Lebesgue constant. Sec. IV analyzes how the apparent deviation can be suppressed.

We elaborate on the role of boundary conditions and delimited expansion domain. In particular, it is argued that both scenarios are intuitively related to the Chebyshev grid. The concluding remarks are given in Sec. V.

## II. THE MATRIX METHOD

In this section, we briefly revisit the algorithm of the matrix method and elaborate on its modification to deal with effective potential with discontinuity. Based on the interpolation associated with a series of the nodes [15], the matrix method rewrites the derivatives of the relevant waveform in terms of their function values on the nodes. Subsequently, the master equation of the black hole QNMs [1]

$$\left[ \frac{\partial^2}{\partial r_*^2} + \omega^2 - V_{\text{eff}} \right] \Psi = 0 , \tag{1}$$

where $V_{\text{eff}}$ is the effective potential, the spatial variable $r_*$ is known as the tortoise coordinates, and the frequency $\omega$ is a complex eigenvalue, can be transformed into a non-standard matrix eigenvalue equation [16, 17],

$$\mathcal{G}\mathcal{F} = 0 , \tag{2}$$

where $\mathcal{G}$ is a $N \times N$ matrix determined by the specific interpolation scheme, and the column matrix $\mathcal{F}$ reads

$$\mathcal{F} = \left( \Psi(x_1), \Psi(x_2), \Psi(x_3), \cdots , \Psi(x_j), \cdots , \Psi(x_N) \right)^T . \tag{3}$$

In practice, the tortoise coordinate $r_*$ is transformed into $x$ whose domain is of finite range, and $\{x_1, \cdots , x_N\}$ are $N$ discrete nodes where the interpolation is performed. One typically choose $x \in [0, 1]$, where the boundaries are located at $x = x_1 = 0$ and $x = x_N = 1$.

To adapt the physical boundary conditions, the asymptotic form at the horizon and spatial infinity is subtracted from the original wave function [18]. As a result of the above process, the boundary conditions of the waveform read

$$\Psi(x = 0) = C_0 \quad \text{and} \quad \Psi(x = 1) = C_1 . \tag{4}$$

Since the asymptotic forms of the wave function have been stripped away, the remaining parts $C_0$ and $C_1$ are constants.

To simply the boundary condition, as proposed in Refs. [16–18], one rewrites $\mathcal{G}$ by introducing the transform

$$\Phi(x) = \Psi(x)x(1 - x) , \tag{5}$$

and rewrites Eq. (2) into

$$\overline{\mathcal{G}}\,\overline{\mathcal{F}} = 0 , \tag{6}$$

where the matrix $\overline{\mathcal{G}}$ is defined by

$$\overline{\mathcal{G}}_{i,j} = \begin{cases} \delta_{i,j}, & i = 1 \text{ or } N \\ \mathcal{G}_{i,j}, & i = 2, 3, \cdots , N - 1 \end{cases} , \tag{7}$$

and

$$\overline{\mathcal{F}} = \left(\Phi(x_1), \Phi(x_2), \Phi(x_3), \cdots, \Phi(x_j), \cdots, \Phi(x_N)\right)^T .\tag{8}$$

Accordingly, the above equation is accompanied by the modified boundary conditions

$$\Phi(x = 0) = \Phi(x = 1) = 0 .\tag{9}$$

The matrix equation Eq. (6) implies that the quasinormal frequencies $\omega$ satisfy

$$\det \overline{\mathcal{G}}(\omega) = 0 ,\tag{10}$$

The roots of Eq. (10) can be found using the standard nonlinear equation solver. The above approach will be denoted by MV1 in the remainder of the manuscript.

As pointed out in [20], the above scheme cannot be straightforwardly applied to the scenario where the effective potential possesses discontinuity. The remedy to the difficulty is to explicitly take into account Israel's junction condition [58, 59] at the point of discontinuity, denoted by the grid points $x = x_c$. To be specific, the wave functions on the two sides of discontinuity are related by [33, 60]

$$\lim_{\epsilon \to 0^+} \left[ \frac{R'(x_c + \epsilon)}{R(x_c + \epsilon)} - \frac{R'(x_c - \epsilon)}{R(x_c - \epsilon)} \right] = \kappa ,\tag{11}$$

and for the master equation given by Eq. (1), one has

$$\kappa = \lim_{\epsilon \to 0^+} \int_{x_c - \epsilon}^{x_c + \epsilon} V_{\text{eff}}(x) dx .\tag{12}$$

In the specific case of a finite jump, the above condition can be further simplified to the vanishing condition of the following Wronskian

$$W(\omega) \equiv R(x_c + \epsilon)R'(x_c - \epsilon) - R(z_x - \epsilon)R'(x_c + \epsilon) .\tag{13}$$

The matrix $\mathcal{G}$ should be revised to adopt the above conditions. As discussed in [20], the modified matrix is *almost* broken into two diagonal sections of block submatrices. The relation given by Eq. (11) or (13) is transformed into a row shared by these two blocks. The above-modified version for discontinuous effective potential will be denoted as MV2 in the remainder of the manuscript.

In the following section, we will show that both schemes presented in this section give reasonable results for black hole QNMs at moderately lower order. However, as one goes to higher orders, Runge's phenomenon emerges and becomes more significant with increasing grid numbers.

## III. RUNGE'S PHENOMENON AND ITS OCCURRENCE IN THE MATRIX METHOD

Named after German mathematician Carl Runge, the phenomenon refers to the oscillation that occurs when using polynomial interpolation at equidistant points to approximate a continuous function. The problem becomes more severe as the degree of the interpolating polynomial increases. Rather than converging to the function being approximated, the polynomial shows increasing oscillations near the endpoints of the interval. It is especially problematic near points where the function to be approximated has discontinuities or sharp turns.

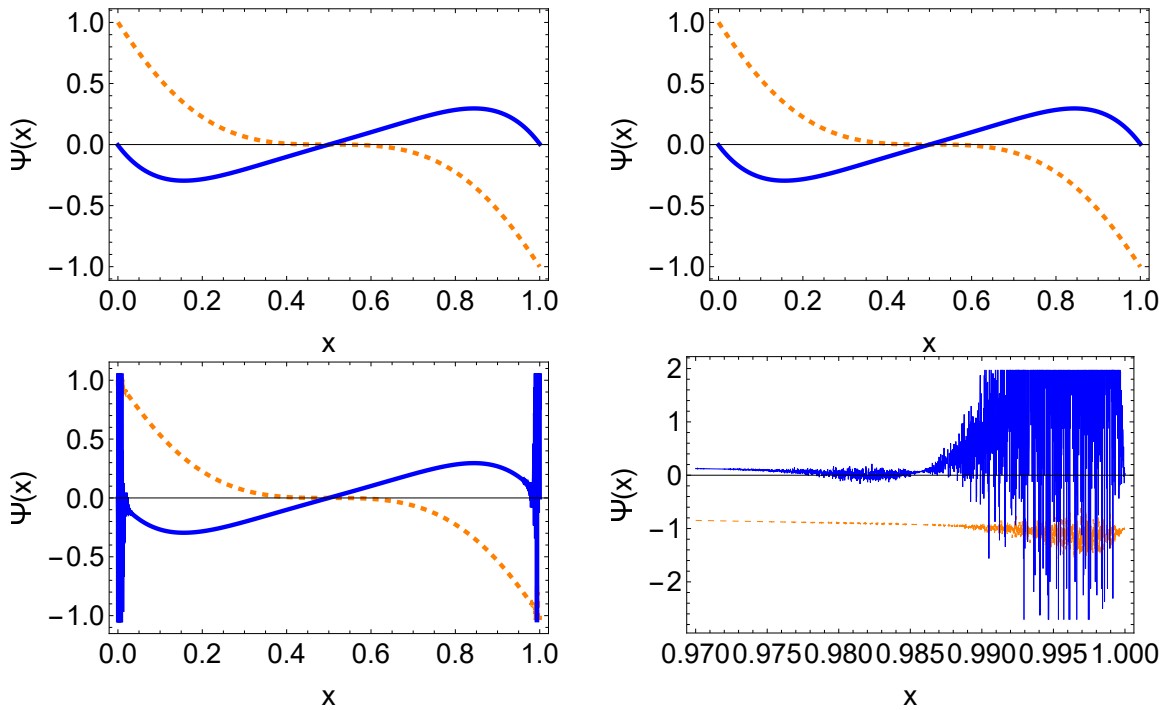

Figure 1. An illustration of Runge's phenomenon for the quasinormal waveform of the fundamental mode for the potential barrier given by Eq. (14). The calculations are carried out for interpolation on a uniform grid with different grid numbers, and the results are represented in the top-left ($n = 10$), top-right ($n = 30$), and bottom-left ($n = 70$) panels. The bottom-right plot shows a zoomed-in portion of the bottom-left panel.

As an illustration, we start with the discussions of the quasinormal waveform of a simple square potential barrier

$$V_{\text{eff}}^0 = V_{\text{barrier}}(x) = \begin{cases} 0 & x < 0 \\ 1 & 0 \leq x \leq 1 \\ 0 & x > 1 \end{cases} . \tag{14}$$

The QNMs of the square potential barrier can be evaluated semi-analytically by identifying the singularities of the reflection and transmission coefficients when viewing the problem as a scattering process against the potential given by Eq. (14). Specifically, the resulting QNMs correspond to the roots of the matrix component $\mathbb{T}_{22} = 0$ [13], which subsequently gives rise to the following equation

$$e^{2i\sqrt{-1+\omega^2}}\left[1 + 2\omega\left(-\omega + \sqrt{-1+\omega^2}\right)\right] + 2\omega\left(\omega + \sqrt{-1+\omega^2}\right) - 1 = 0 . \tag{15}$$

The fundamental mode corresponds to the root of Eq. (15) with the smallest magnitude for the imaginary part, which is found to be

$$\omega_0 = 4.63002044069 - 5.218929351124i , \tag{16}$$

and the corresponding waveform possesses the analytic form,

$$\Psi(x) = \frac{\sqrt{\omega_0^2 - 1} - \omega_0}{\sqrt{\omega_0^2 - 1} + \omega_0} e^{i\sqrt{\omega_0^2-1}x - 2i\sqrt{\omega_0^2-1}} + e^{-i\sqrt{\omega_0^2-1}x} , \tag{17}$$

up to an irrelevant normalization constant.

To proceed, we perform an interpolation of the waveform Eq. (17) using a uniform grid and show the resultant polynomial in Fig. 1 The calculations are carried out for different grid numbers $n = 10, 30$, and 70. It is observed that for small grid numbers, the interpolated waveforms are essentially identical, as shown in the top-left ($n = 10$) and top-right ($n = 30$) panels. However, for a more significant grid number, roughly $n \gtrsim 60$, significant oscillations appear at the edges of the interval, indicating the onset of Runge's phenomenon. This is demonstrated by the bottom-left and bottom-right panels. The above results indicate that Runge's phenomenon may play a role in the matrix method when interpolation is performed on a uniform grid. As will be elaborated further below, the relevant grid number when the original approach ceases to provide reasonable results is numerically consistent with the above results.

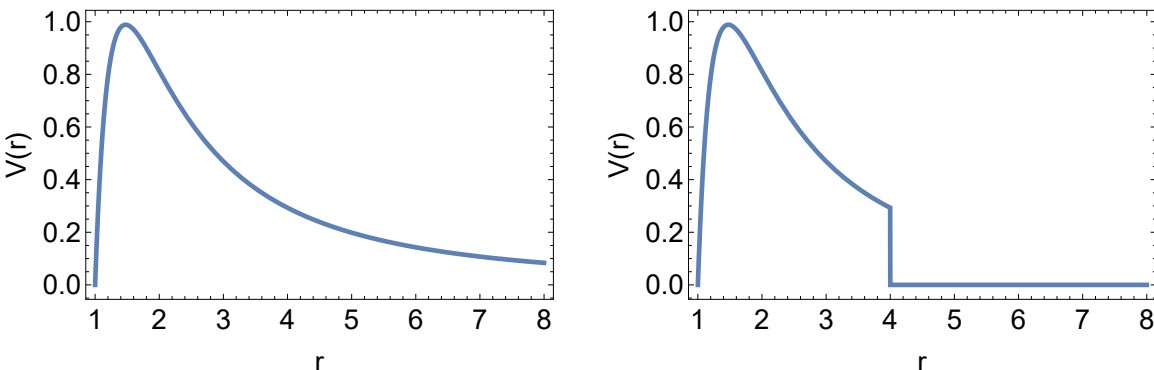

Figure 2. The effective potentials adopted in the present study for axial gravitational perturbations in Schwarzschild black hole metric. Left: the Regge-Wheeler potential defined in Eq. (18). Right: The truncated potential Eq. (19), where a cut is implemented at $r_c = 4$.

In what follows, we explore the occurrence of Runge's phenomenon numerically regarding the two versions of the matrix method discussed in the last section. Although reasonable results are retrieved under moderate circumstances, we show that Runge's phenomenon does become a severe problem at higher orders.

Concerning the purpose of the present study, we consider two specific forms of effective potential as follows. The first one is the Regge-Wheeler potential for Schwarzschild black holes, which reads

$$V_{\text{eff}}^1 = V_{\text{RW}}(r) \equiv \left(1 - \frac{r_h}{r}\right)\left[\frac{\ell(\ell + 1)}{r^2} + (1 - s^2)\frac{r_h}{r^3}\right], \tag{18}$$

where $r$ is the radial coordinate, related to the tortoise coordinate by $r_* = \int \frac{dr}{1 - \frac{r_h}{r}}$, $r_h$ is the location of the horizon, and $\ell$ corresponds to the angular momentum. The variable $x$ is defined as $x = \frac{r - r_h}{r}$. The form of the potential is shown in the left plot of Fig. 2.

Second, for the case of effective potential with discontinuity, we consider the following form obtained by truncating the Regge-Wheeler potential Eq. (18) at $r = r_c$. As shown in the right plot of Fig. 2, it is defined by

$$V_{\text{eff}}^2 = \begin{cases} V_{\text{RW}} & r \le r_c \\ 0 & r > r_c \end{cases}. \tag{19}$$

In the remainder of this paper, calculations are carried out for the fundamental mode of axial gravitational perturbations with $s = -2$, $r_h = 1$, and $\ell = 2$. Nevertheless, the conclusion drawn

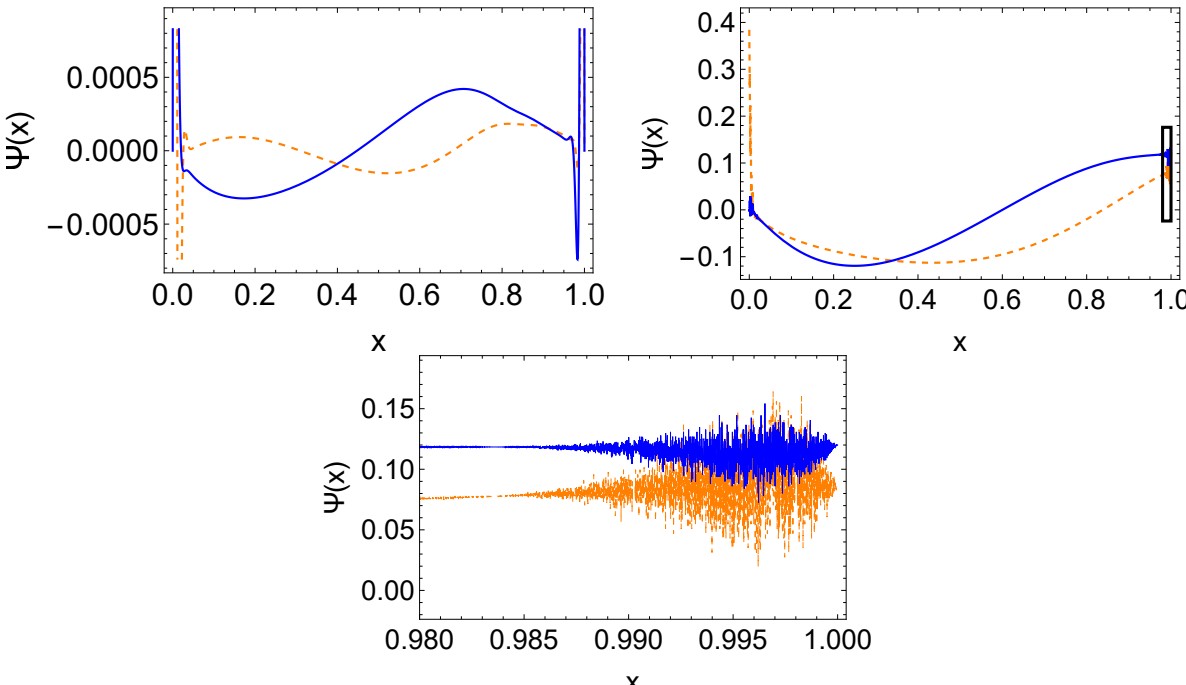

Figure 3. The Runge phenomenon is demonstrated by significant oscillations in the interpolated wave-forms, occurring at both edges of the interval $x = 0$ and 1. The real and imaginary parts of the waveform $\Psi(x)$ are shown by the dashed orange and solid blue curves. Top left: The evaluated waveform of the method MV1, calculated for the effective potential shown in the left plot of Fig. 2 using 61 grid points. Top right: The evaluated waveform of the method MV2, calculated for the effective potential shown in the right plot of Fig. 2, also using 61 grid points. Bottom: A zoomed-in portion of the top right plot, indicated by the black square.

from axial gravitational perturbations has also been verified to be, by and large, valid to other types of perturbations with different angular momenta. As a reference for the accurate value, the quasinormal frequency for the effective potential Eq. (18) is also evaluated by employing the continued fraction method [14] at 300th order. The value is found to be

$$\omega_{\text{CF}} = 0.74734336883598689863 - 0.17792463137781263197i.$$

The quasinormal frequency for the effective potential Eq. (19) can also be evaluated by using a recurrence Taylor expansion scheme [61] at 300th order, which reads

$$\omega_{\text{RTE}} = 0.79425298413668122287 - 0.14836993024971837407i.$$

The evaluated QNM frequencies for the effective potentials Eqs. (18) and (19) are presented in Tabs. I and II. In Tab. I, as the grid number increases, it is observed that the resulting complex frequency first converges and then diverges as the grid number goes beyond 41. Compared to the value from the continued fraction method, at $N = 35$, the method MV1 attains the highest significant figures $N_{\text{Sig}} = 8$. In the top left plot of Fig. 3, we show the corresponding waveform for the grid number $N = 61$. Even though the boundary condition has enforced $\Psi(0) = \Psi(1) = 0$, significant oscillations appear near the edges, which is a typical demonstration of the Runge's phenomenon [57].

Table I. The calculated fundamental mode employing the matrix method (MV1) for the Regge-Wheeler effective potential Eq. (18) by using different grid numbers.

| Grid number $N$ | $\omega$ | $\Lambda_n/\Lambda_{35}$ |
| --- | --- | --- |
| 11 | $0.74736508236138663485 - 0.17788604312667105354i$ | $2.6 \times 10^{-7}$ |
| 15 | $0.74733828947826767365 - 0.17792416187562642623i$ | $2.8 \times 10^{-6}$ |
| 21 | $0.74734366219031232799 - 0.17792481061688616289i$ | $1.2 \times 10^{-4}$ |
| 25 | $0.74734340368062390603 - 0.17792456992234014614i$ | $1.5 \times 10^{-3}$ |
| 31 | $0.74734336062217307506 - 0.17792463238941793080i$ | $7.3 \times 10^{-2}$ |
| 35 | $0.74734336866050136697 - 0.17792463358986663649i$ | $1$ |
| 41 | $0.71366355617972504354 - 0.34217275143274520441i$ | $52$ |
| 51 | $0.65235499436784095031 - 0.27756660050373268323i$ | $4.1 \times 10^{4}$ |
| 61 | $0.66424482851821401498 - 0.24986341753963809900i$ | $3.4 \times 10^{7}$ |

Table II. The calculated fundamental mode employing the modified matrix method (MV2) for the Regge-Wheeler effective potential Eq. (19) by using different grid numbers.

| Grid number $N$ | $\omega$ | $\Lambda_n/\Lambda_{41}$ |
| --- | --- | --- |
| 11 | $0.78332877298546168772 - 0.15890873993628376931i$ | $5.0 \times 10^{-9}$ |
| 15 | $0.79164131485624724313 - 0.14726475130268517427i$ | $5.3 \times 10^{-8}$ |
| 21 | $0.79427451740740686113 - 0.14827048584492279671i$ | $2.2 \times 10^{-6}$ |
| 25 | $0.79425867578740535325 - 0.14836310303749736941i$ | $2.8 \times 10^{-5}$ |
| 31 | $0.79425333074579068309 - 0.14836965173112674700i$ | $1.4 \times 10^{-3}$ |
| 35 | $0.79425305864517265298 - 0.14836991724415013199i$ | $1.9 \times 10^{-2}$ |
| 41 | $0.79376170367437759294 - 0.14887545803671615841i$ | $1$ |
| 51 | $0.81212553473007960577 - 0.06803874222674382039i$ | $7.8 \times 10^{2}$ |
| 61 | $0.59169792910015569757 - 0.35049028115298366817i$ | $6.5 \times 10^{5}$ |

Similarly, the results obtained by the method MV2 for the effective potential Eq. (19) are shown in Tab. II. Reasonable results for black hole QNMs are observed at moderately lower order $N \leq 35$, with an agreement of $N_{\mathrm{Sig}} = 6$ significant figures. As one goes to higher orders, again, Runge's phenomenon emerges and becomes more significant with increasing grid numbers. The corresponding waveform for $N = 61$ is presented in the top right and bottom plots of Fig. 3. As clearly shown in the zoomed-in shot near the right edge $x = 1$, the oscillations have undermined the method's precision.

In literature, the undesirable deviations associated with Runge's phenomenon can be estimated in terms of the rapid growth of the Lebesgue constant. For a given set of grid points on the interval $[a, b]$, the Lebesgue constant is defined as the norm of a linear operator associated with the interpolation process based on the grid. The process in question is essentially a projection $\mathcal{X}$ from the space $C([a, b])$ of all continuous functions on the relevant interval $[a, b]$ onto the subspace $\Pi_n$ of polynomials up to degree $n$. As an operator norm, the definition of Lebesgue constant resides on the definition of the norm on $C([a, b])$, the space of continuous functions on the interval. It can

be shown that the Lebesgue constant is the maximum value of the Lebesgue function

$$\lambda_n(x) = \sum_{j=0}^{n} \left| \ell_j(x) \right|, \tag{20}$$

over the domain, i.e.,

$$\Lambda_n = \sup_{x \in [a,b]} \lambda_n(x), \tag{21}$$

where $\ell_j$'s are the Lagrange polynomials

$$\ell_j(x) = \prod_{i=0, i \neq j}^{n} \frac{x - x_j}{x_i - x_j}. \tag{22}$$

Intuitively, the Lebesgue constant bounds the interpolation error and provides a quantitative measure of how close the interpolant of a function is with respect to the best polynomial approximant of the function of the same degree. One may use the Weierstrass approximation theorem to address the remainder in the Lagrange interpolation formula. Two factors govern the upper bound of the latter: the nodal function and the $(N + 1)$th derivative of the waveform [62]. In the case of a uniform grid, the Lebesgue constant can be estimated according to Turetski [63]. In the last columns of Tabs. I and II, the Lebesgue constants are given with respect to the order in which the divergence starts to emerge.

## IV. RECIPES FOR ACCURACY IMPROVEMENT

The last section shows that Runge's phenomenon poses a severe challenge to the accuracy of the matrix method at higher orders. In the present section, we investigate possible remedies to the problem and present numerical evidence for the elaborated algorithms. Moreover, we explore the connection between the improvements of the matrix method and the well-known Chebyshev grid.

In literature, one of the most well-known approaches to suppress the Runge phenomenon is the Chebyshev grid [57]. Exponential convergence can be achieved by minimizing the Lagrange interpolation error [62], specifically, the maximum of the nodal functions. The resulting Chebyshev grid (of the second kind) is given by

$$y_j^{\text{Cheb}} = \cos\left(\frac{\pi j}{N}\right), \quad j = 0, 1, \cdots N. \tag{23}$$

It is featured by non-uniform distribution, for which the nodes cluster at the two edges of the interpolation interval $[-1, 1]$. The Fekete grid adopts a similar strategy and maximizes the Vandermonde determinant, yielding a smaller Lebesgue constant. Also, a few approaches based on equidistant nodes on different basis have been proposed [64]. Notably, most proposed schemes involve a specific nonlinear transform. The latter introduces either a non-uniform grid distribution or sophisticated regularization factors for the expansion coefficients. Regarding the matrix method, implementing such processes often undermines the analytic form of the matrix $\mathcal{G}$, which, in turn, potentially leads to a price in terms of either computational time or precision. Moreover, it was pointed out [65] that the spatial dependence of a differential equation owing to the Chebyshev discretization could be undesirable as it might become very *stiff*. In what follows, we elaborate on two possible modifications to the matrix method that effectively suppress Runge's phenomenon.

### IV.1. Suppression by boundary conditions

However, while maintaining a uniform grid, it can be shown that the convergence of the matrix method is manifestly achieved in numerical practice. A better convergence can be attained by lifting the boundary condition enforced by Eq. (9). Specifically, one resorts to the original matrix equation Eq. (2) for effective potential without discontinuity. In other words, one solves for the quasinormal frequencies $\omega$ by

$$\det \mathcal{G}(\omega) = 0 \, , \tag{24}$$

where the corresponding column matrix $\mathcal{F}$ constitutes the wave function $\Psi(r_*)$ at the grid points. The above recipe with enforced boundary conditions will be denoted as MV3.

The corresponding quasinormal frequencies evaluated for the effective potential Eq. (18) using the above approach are presented in Tab. III. By comparing to those given in Tab. I obtained using MV1, the convergence of the method MV3 is manifestly shown. Moreover, when compared to the value from the continued fraction method, the method MV3 achieves $N_{\text{Sig}} = 11$ significant figures. In the left plot of Fig. 4, we show the corresponding waveform for the grid number $N = 61$. It is observed that the enforced boundary condition $\Psi(0) = \Psi(1) = 0$ is lifted, and no noticeable oscillation is observed.

Table III. The calculated fundamental mode employing the matrix method (MV3) for the Regge-Wheeler effective potential Eq. (18) by using different grid numbers.

| Grid number $N$ | $\omega$ |
|---|---|
| 11 | $0.74734142569718546206 - 0.17792359012679884957i$ |
| 15 | $0.74734339428402995916 - 0.17792493160929834949i$ |
| 21 | $0.74734337435074057974 - 0.17792461580026767347i$ |
| 25 | $0.74734336575997473616 - 0.17792463033137827244i$ |
| 31 | $0.74734336894429315299 - 0.17792463171019160783i$ |
| 35 | $0.74734336892429329525 - 0.17792463136858928592i$ |
| 41 | $0.74734336883626139147 - 0.17792463137856964651i$ |
| 51 | $0.74734336883626139147 - 0.17792463137856964651i$ |
| 61 | $0.74734336883609414984 - 0.17792463137782069136i$ |

### IV.2. Suppression by delimited expansion domain

For the case where discontinuity is present, a generalized matrix method was proposed to suppress the Runge phenomenon [21]. The modified algorithm does not perform a full-rank interpolation using the entire $N$ nodes. Instead, for a given grid point $x_i$, only a subset of $P$ is utilized. The latter governs the polynomial order of such a "localized" interpolation and satisfies [66]

$$P < \sqrt{\frac{1}{\chi}} \sqrt{N}, \tag{25}$$

where

$$\chi > \frac{2}{\pi^2}. \tag{26}$$

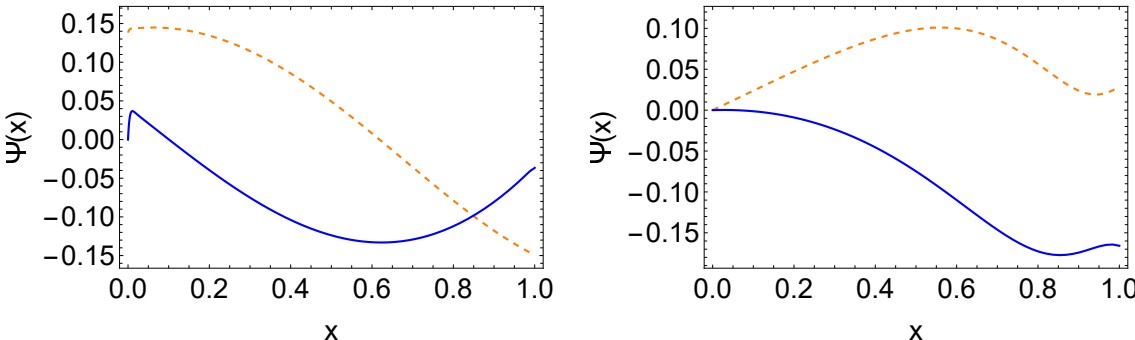

Figure 4. The waveforms $\Psi(x)$ are obtained by the approaches discussed in the present section using 61 grid points. The real and imaginary parts are shown by the dashed orange and solid blue curves. Left: The evaluated waveform of the method MV3, calculated for the effective potential shown in the left plot of Fig. 2. Right: The evaluated waveform of the method MV4, calculated for the effective potential shown in the right plot of Fig. 2.

Such a recipe can be readily applied to the case of effective potential with discontinuity Eq. (19) discussed in Sec. III. As a result, the matrix $\bar{\mathcal{G}}$ becomes more sparse, as only $P$ elements will be filled on each row. Subsequently, the calculation becomes more efficient when compared with its counterpart with an identical grid number. Typically, a more significant value of $N$ can be adopted at moderate computational time, resulting in better performance. This modification with delimited expansion domain will be referred to as MV4.

The corresponding quasinormal frequencies evaluated for the effective potential Eq. (18) using the above approach are presented in Tab. IV. By comparing with those given in Tab. II obtained using MV2, the convergence of the method MV4 is significantly improved. In particular, when compared to the value from the continued fraction method, the method MV4 obtains $N_{\text{Sig}} = 11$ significant figures. In the right plot of Fig. 4, we show the corresponding waveform for the grid number $N = 265$. It is observed that there is no oscillation at the boundary of the interval. Besides increasing precision, the method MV4 also warrants satisfactory efficiency.

Table IV. The calculated fundamental mode employing the modified matrix method (MV4) for the Regge-Wheeler effective potential Eq. (19) by using different grid numbers.

| Grid number $N$ | Polynomial $P$ | $\omega$ |
|---|---|---|
| 33 | 8 | $0.79427102360997376129 - 0.14815202532364477392i$ |
| 61 | 10 | $0.79425457798200380894 - 0.14837036928039701020i$ |
| 85 | 12 | $0.79425298430556435132 - 0.14836995464695804226i$ |
| 113 | 14 | $0.79425298391364355806 - 0.14836993034017380308i$ |
| 145 | 16 | $0.79425298413593529556 - 0.14836993024928454064i$ |
| 181 | 18 | $0.79425298413719213174 - 0.14836993025040434318i$ |
| 220 | 20 | $0.79425298413719036408 - 0.14836993025041291405i$ |

### IV.3.   Connection with Chebyshev grid

The delimited expansion approach Eqs. (25) discussed in the previous subsection was shown to be particularly useful for higher-order calculations. We argue that the approach is in accordance with the spirit of the Chebyshev grid. In particular, it can be viewed as inspired by the *overdetermined least-square method* [64, 66] and *mock-Chebyshev* grid [66, 67]. Both approaches were first introduced by Boyd and Xu, based on the findings of Rakhmanov [68].

The overdetermined least-square method [64, 66] is a generalization of the standard linear regression. Intuitively, if the physical law (after the appropriate transformation of the relevant variables) is known to be linear, then linear regression should be used even though the data size is significant. In other words, any attempt to fit the data with a higher-order polynomial will result in overfitting. According to the overdetermined least-square method, one should perform the least square fit to a polynomial, whose degree $P$ is typically much smaller than the size of the data $N$.

From a mathematical perspective, the above conclusion is based on Rakhmanov's theorem (theorem six of [66]). Let us consider a scenario when the data is only available on a uniformly distributed grid with a total of $N$ nodes. By employing a full-rank polynomial fit, Runge's phenomenon will likely occur and lead to undesirable outcomes. Instead, one considers a polynomial fit of a minor rank $P$ with $P < N$. The idea behind the proposed model is to estimate the upper bound of the deviation between the fit and the original (unknown) function. This can be done by bringing in the Chebyshev interpolant, which will be interpolated on the Chebyshev grid (the function values on which are unknown). However, the deviation can be estimated in terms of the deviation of the Chebyshev interpolant (theorem five of [66]). As it turns out, the deviation is roughly proportional to $\sqrt{N}$. When used in conjunction with Rakhmanov's theorem, it leads to the conclusion that a polynomial fit of smaller $P$ is more favorable, giving rise to convergent results as $N \to +\infty$. More quantitatively, the Runge region, an area defined by error isosurface inside of which any pole of the target function implies Runge's phenomenon, shrinks as the ratio between the polynomial degree and grid points decreases (theorem seven of [66]).

Alternatively, the mock-Chebyshev method chooses to perform a polynomial interpolation of degree $P$ using a given number of $P$ nodes. The approach can be understood intuitively in terms of the Chebyshev grid. Specifically, the grid points' clustering near the end of the interpolation interval effectively suppresses the significant amplitude oscillations in the interpolant. In particular, the nodes' density is quadratic in the total grid number $N$, in accordance with Rakhmanov's findings. The latter states [68] that convergence can be restored for the function's singularity close to the interval if and only if the number of grid points $N$ grows at least as fast as the square of the polynomial degree $P$. Motivated by the above considerations, the authors of [66] proposed to choose $P$ points out of an equispaced grid containing $N$ nodes, which are closest to the Chebyshev grid while satisfying Eq. (25). While such a choice is intuitive, it is apparent that the generalized matrix method with delimited expansion region follows a very similar spirit.

On the other hand, the release of the enforced boundary condition Eq. (9) corresponds to multiplying the factor $x(1 - x)$. The latter is readily rescaled to the relevant interval of $[-1, +1]$ for the Chebyshev grid by transforming $y = 2x - 1$. Compared with the sampling of Chebyshev grid Eq. (23), it becomes apparent that the above factor in the matrix method plays the role of cos function in the Chebyshev grid, which effectively leads to a clustering of nodes near the edges. We conclude that both versions of the generalized matrix method have their origin in the Chebyshev grid.

## V. CONCLUDING REMARKS

As pointed out by Runge more than a century ago, higher degree polynomial interpolation in terms of a uniformly spaced grid might lead to an undesirable convergence problem, known as Runge's phenomenon. In this regard, the matrix method for black hole QNMs demonstrates surprising robustness in practice. The present paper aims to explore this aspect and provide a feasible explanation. For most metrics, we showed that an appropriate choice of boundary conditions gives rise to desirable suppression of Runge's phenomenon. In cases where discontinuity is present in the effective potentials, it is shown that a generalized version of the method with a delimited expansion domain leads to satisfactory results. We argue that both scenarios are closely related to or practically imitating the Chebyshev grid. Although the implementations are somewhat distinct, the matrix method with delimited expansion domain is closely associated with the overdetermined least-square and mock-Chebyshev methods proposed earlier in the applied mathematics literature. One concludes that the matrix method is a sound and helpful tool for the numerical study of black hole QNMs.

Implementing the matrix method is relatively straightforward, chiefly due to the uniform grid used across all of its variations. This approach improves computational efficiency, as adopting a non-uniform grid like the Chebyshev one leads to a more complex differential matrix. Additionally, round-off errors are inevitably introduced due to the inclusion of trigonometric functions. Nonetheless, it is worth investigating whether the matrix method can be adapted to incorporate the Chebyshev grid and the spectral method. The primary challenge for the latter is expected to lie in computational cost, as the grid points often do not take rational values. We plan to explore these topics in the near future.

### ACKNOWLEDGMENTS

We acknowledge insightful discussions with Kai Lin. This work is supported by the National Natural Science Foundation of China (NNSFC). We also gratefully acknowledge the financial support from Fundação de Amparo à Pesquisa do Estado de São Paulo (FAPESP), Fundação de Amparo à Pesquisa do Estado do Rio de Janeiro (FAPERJ), Conselho Nacional de Desenvolvimento Científico e Tecnológico (CNPq), Coordenação de Aperfeiçoamento de Pessoal de Nível Superior (CAPES), A part of this work was developed under the project Institutos Nacionais de Ciências e Tecnologia - Física Nuclear e Aplicações (INCT/FNA) Proc. No. 464898/2014-5. This research is also supported by the Center for Scientific Computing (NCC/GridUNESP) of São Paulo State University (UNESP).

---

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
