# Peer review of "Matrix method and the suppression of Runge’s phenomenon"

_SciPost Physics Core_

## Round 1 · Referee Report · Anonymous (Referee 1) · 2024-3-12

Strengths

The article successfully addressed an important technical aspect of the matrix method related to the Runge's phenomenon.

Weaknesses

There is an extensive discussion about the method, the QNMs and the somehow of academic only interesting "pseudospectra". But even thought the article sited dozens of publications it missed some of the very first articles using the matrix method. Actually, in computational physics this is typically referred as "Numerov's method" .

Report

This article is part of a series written by some of the authors over an 8-year time period. They were successful in promoting the use of the matrix method for studying quasi-normal modes. In the article, they demonstrated the potential of this method in addressing the issue of Runge's phenomenon. This result alone is sufficient to make the paper interesting to readers. However, the authors acknowledge that there is currently no solution for cases where the effective potential is discontinuous. I consider this to be a minor weakness since these types of potentials have limited importance for astrophysics and the instabilities reported are mostly of academic interest. Overall, the article presents original research and, in my opinion, meets the criteria for publication in the journal.

Requested changes

While reading the review article by Kokkotas & Schmidt in Living Reviews (1999), I came across the first ever uses of the matrix method in calculating QNMs:
** of black holes:
-- Bhabani Majumdar and N. Panchapakesan PRD, 40, 2568 (1989)
-- E.W. Seidel PRD 41, 2986 (1990)
** of neutron stars
-- K.D. Kokkotas and J. Ruoff A&A 365, 565 (2001)
-- S. Boutloukos and H-P. Nollert PRD 75, 043007 (2007)

I recommend to include these characteristic references in the introduction.

---

## Round 1 · Referee Report · Anonymous (Referee 2) · 2024-4-12

Report

The matrix method is widely recognized as an effective approach for computing Quasinormal Modes (QNMs). Typically, this method discretizes the eigenvalue problem using a grid of uniformly spaced points and computes the system’s eigenvalues, which are the QNMs. However, employing polynomial interpolation at these grid points can often invoke Runge’s phenomenon, leading to errors in approximating eigenstates. This study focuses on the occurrence of Runge’s phenomenon when utilizing the matrix method to calculate QNMs, proposing that the introduction of suitable boundary conditions can significantly mitigate this issue. The work also examines cases with discontinuous effective potentials. Though the manuscript seems to be suitable for publication in scipost, I suggest certain revisions for improvement.

Key Concern:
The authors have identified worsened convergence for QNMs with an increasing number of grid points. However, after selecting appropriate boundary conditions, Runge’s phenomenon is suppressed. The authors link the convergence of QNMs with Runge’s phenomenon. Since Runge’s phenomenon typically emerges during interpolation, it is imperative to ascertain whether the enhancement in convergence can be directly ascribed to its mitigation. The authors are encouraged to furnish substantial evidence or a compelling argument to substantiate this link.

Secondary Concerns:

There is inconsistent use of symbols to denote the number of grid points; ‘N’ is predominantly used, yet ‘n’ appears in several instances (e.g., page 7, Figure 1, Table 1, and Table 2). A uniform notational convention should be applied throughout.
The symbol (z_x) featured in Equation 13 has not been previously introduced. It is requested that the author confirm whether this is a typographical error or, if the symbol has a different meaning, to provide an explanation.
There is a punctuation oversight on page 7: “To proceed, we perform an interpolation of the waveform Eq. (17) using a uniform grid and show the resultant polynomial in Fig. 1 The calculations are conducted for various grid numbers n = 10, 30, and 70,” where a punctuation mark is absent between Fig. 1 and The.

Recommendation

Ask for minor revision

---

## Editorial Decision

resubmitted